# Health-related quality of life among women aging with and without HIV in Peru

Diego M. Cabrera [1,2], Meibin Chen[1], Mijahil P. Cornejo[3], Yvett Pinedo[4], Patricia J. Garcia[2], Evelyn Hsieh[1]*

1 Section of Rheumatology, Allergy and Immunology, Yale School of Medicine, New Haven, Connecticut, United States of America, 2 Epidemiology, STD, and HIV Unit, School of Public Health, Universidad Peruana Cayetano Heredia, Lima, Peru, 3 Department of Rheumatology, Hospital Nacional Arzobispo Loayza, Lima, Peru, 4 Department of Infectious Diseases, Hospital Nacional Arzobispo Loayza, Lima, Peru

* evelyn.hsieh@yale.edu

## Abstract

### Background

Studies have shown that women aging with HIV have significantly lower health-related quality of life (HRQoL) compared to women without HIV. However, no studies have examined this issue in Latin America and the Caribbean. We aimed to explore HRQoL measured by the 36-Item Short Form Health Survey (SF-36) among women aging with and without HIV in Peru.

### Materials and methods

We conducted a cross-sectional study at a large HIV-clinic in Peru. Outcomes of the SF-36 were evaluated, exploring the relationship between physical activity (International Physical Activity Questionnaire), sociodemographic factors (ethnicity, alcohol/tobacco use, age, BMI) and clinical data (AIDS progression, treatment duration, CD4+ cell count and viral load, years since HIV diagnosis) with HRQoL using regression analysis. Statistical significance was set with a two-tailed p-value <0.05.

### Results

We enrolled 427 women (175 HIV-infected) with mean age of 54±8 years. From the SF-36 individual domains: physical functioning, role limitations due to physical and emotional health, and emotional wellbeing were significantly lower for HIV-infected women. Summary component scores were lower for the HIV-subset for both physical (45.8 vs 47.3) and mental (45.1 vs 45.8) components, although they did not achieve statistical significance. Regression analysis of the HIV-infected women revealed that the physical component score was significantly associated with physical activity, ethnicity, and chronic comorbidities while the mental component was significantly associated with physical activity, employment, and CD4 + cell count.

### Conclusion

In our study, HIV-infected women scored lower in both physical and mental component scores. Important determinants for each component included CD4+ cell count as an

**Data Availability Statement:** All datasets are available from the Open Science Framework public repository (https://osf.io/hfg64/).

**Funding:** DMC serves as a Fogarty Global Health Trainee and is supported by the Fogarty

International Center (FIC, URL: https://www.fic.nih.gov/) at the National Institutes of Health (NIH) and the National Institute of Arthritis and Musculoskeletal and Skin Diseases (NIAMS) under grant number D43TW010540. EH is supported by FIC. The funders had no role in study design, data collection and analysis, decision to publish, or preparation of the manuscript.

**Competing interests:** The authors have declared that no competing interests exist.

**Abbreviations:** HRQoL, Health-related quality of Life; SF-36, 36-Item Short Form Health Survey; IPAQ, International Physical Activity Questionnaire; PLWH, People living with HIV; ART, Antiretroviral treatment; HNAL, Hospital Nacional Arzobispo Loayza; MET, Metabolic Equivalent of Task; MOS, Medical Outcomes Study.

assessment of HIV severity for the mental component, and ethnicity, reflecting socio-cultural factors, for the physical component. These results reveal the importance of a holistic approach to addressing HRQoL in this population. Better understanding of these factors will help shape future policies and interventions to improve HRQoL of women aging with HIV.

## Background

In the past two decades, the introduction of antiretroviral therapy (ART) has changed HIV from a fatal disease to a chronic condition that can be managed with long-standing medication. However, people living with HIV (PLWH) remain at increased risk for developing chronic conditions associated with aging, such as neurocognitive impairment, cardiovascular disease, as well as decreased physical functioning [1]. Reflecting this, PLWH continue to have significantly lower health-related quality of life (HRQoL) than the general population [2]. Data suggest that in addition to the burden of living with chronic infection, PLWH are more likely to be adversely affected by changes in their environment and in their social background [3].

Compared to general quality of life, HRQoL is more stringently defined as the way health is estimated to affect an individual's quality of life, or the way an individual perceives their own well-being with regards to physical, mental and social domains of health [4]. Measures of HRQoL in individuals living with chronic conditions such as HIV allows for the evaluation of the efficacy of certain medications as well as for exploring the influence of HIV and associated comorbidities [2, 5]. The 36-Item Short Form Health Survey (SF-36) is a widely used, validated instrument for HRQoL evaluation [6]. This tool has proved to be useful in estimating the burden of different conditions on HRQoL among many populations [7]. The use of the SF-36, among other instruments, to evaluate HRQoL in HIV-infected populations has been reported in upper-income settings [8–10] and a few other Latin American countries [4, 6, 11], but not in Peru.

Women represent an estimated 31% of adults living with HIV in Latin America. Their vulnerability to HIV is related to many diverse influences, such as social, economic and policy-related factors [12]. Throughout the world, the unequal social status of women places them at higher risk for contracting HIV with reported barriers to access for prevention and treatment programs [13]. Furthermore, they may experience more HIV-related stigma compared with men due to their generally devalued status and the patriarchal power dynamics evidenced in certain cultures, which is highly prevalent in Latin America [14]. It is not uncommon in Peru for women to be expected to maintain culturally defined moral traditions and to adhere to conservative gender roles that are not conducive to open conversations related to sexuality. Valencia-Garcia *et al*. described that Peruvian women are often perceived to be the unfaithful partner when diagnosed with HIV, and thus experience stigma within society [15]. In Peru, as of 2018 there were at least 18,000 women living with HIV, representing 23% of all PLWH. Among them, 78% have received treatment [16] through the Peruvian National HIV Program since 2004 [17]. Most studies and interventions for women with HIV in Latin America have focused on reproductive and sexual health of pregnant women and sex workers [12] but have rarely focused on longer-term physical and mental health needs of middle-aged and older women living with HIV.

We therefore designed a study to examine health-related quality of life, as measured by the SF-36 among a group of middle-aged and older women with HIV compared with a group of women without HIV in Lima, Peru. We evaluated factors associated with HRQoL outcomes in both groups, including physical activity using the International Physical Activity

Questionnaire (IPAQ), another well-validated tool, and hypothesized that HIV status may be an important predictor for physical and mental health of this population.

## Materials and methods

### Study design and participants

We conducted a hospital-based observational cross-sectional study. Participants were recruited at the HIV-clinic of Hospital Nacional Arzobispo Loayza (HNAL), a large public hospital in Lima, Peru for a study regarding musculoskeletal health between October 2019 and March 2020. We consecutively invited HIV-positive women aged ≥40 years who presented for routine care visits at HNAL during the study period, and concurrently recruited a control group of HIV-uninfected women aged ≥40 years. We used two strategies to recruit women in the control group, including inviting community-dwelling women from surrounding neighborhoods as well as advertising the study among hospital staff (nurses, office administrators assistants and technicians). Exclusion criteria for both groups included primary residency outside of Lima, pregnancy, history of or treatment for osteoporosis, and illiteracy.

### Measures and data collection

At the time of enrollment, all participants completed a 15-minute self-administered questionnaire that included sociodemographic characteristics, the SF-36, and the IPAQ. Both tools were piloted among 10 patients and 10 clinic staff before recruitment began to confirm that questions were easily understandable and culturally appropriate. The questionnaire was administered to all patients by a single study staff member (D.M.C.) to avoid misinterpretation of the questions and to resolve any uncertainties. HIV-related data were obtained from clinical records, and viral suppression was defined as a current viral load count ≤50 copies.

### Sociodemographic and clinical characteristics

Background information was obtained regarding: personal characteristics including age, self-reported ethnicity (White, Mestizo [mixed ethno-racial ancestry] and other [Black, Andean, Nikkei]), educational status (less than high school vs. high school and beyond), occupation (employed vs unemployed), marital status (single/divorced/separated/ widowed vs married/ cohabitant), smoking history (ever vs never), current alcohol status (yes vs no); HIV-related factors such as nadir and current CD4+ cell count, nadir and current viral load, time since diagnosis, ART duration, history of AIDS diagnosis (yes vs no, with AIDS status defined by CD4+ cell count less than 200 cell/$m^3$); and other clinical characteristics such as body mass index (BMI), hip and waist circumference, menopausal status, and other chronic comorbidities (hypertension, type 2 diabetes, cardiovascular disease, chronic renal disease, mental health illness, chronic respiratory disease, others).

### Measure of physical activity (IPAQ)

The IPAQ was created as an instrument for cross-national assessment of physical activity and for standardization of measures of health-related physical activity behaviors across different sociocultural contexts [18]. We used the short (7-item) version that has been validated in multiple languages, including Spanish, in a study measuring the relation and differences between moderate-to-vigorous physical activity obtained by the IPAQ on two occasions, finding significantly correlated measures (r = 0.55, p<0.01) [19]. The IPAQ uses seven questions to explore the amount of time spent on vigorous-intensity activity, moderate-intense activity, walking and sitting during the past seven days. Answers collected for each category were weighted to

calculate a total weekly metabolic equivalent of task (MET) score in minutes to obtain the total energy cost of the population's activity on those seven days [20]. Categorical labels of low, moderate, and high levels of physical activity were assigned according to official scoring guidelines from the IPAQ [21]. We used the IPAQ as an additive to further explore physical activity assessment within the spectrum of the physical component of the SF-36.

## SF-36

The SF-36 is a widely adopted tool used in clinical research to measure HRQoL that has been validated in many languages including Spanish, in a study within 2,144 subjects from Chile with a high internal consistency scale (Cronbach's alpha 0.86–8.87), [22–24]. This questionnaire was developed as a product of the Medical Outcomes Study (MOS) [17] and yields eight domains of evaluation: physical functioning, role limitations due to physical health, role limitations due to emotional problems, energy/fatigue, emotional well-being, social functioning, pain, and general health. Domain scores were calculated according to the RAND Corporation [25] "RAND 36-item Health Survey 1.0", which transforms scores to a scale of 0–100 in which higher scores indicate a "better" HRQoL [10]. These domains were further aggregated into two summary measures: the Physical Component Summary score (physical component) and the Mental Component Summary score (mental component). To calculate summary scores, SF-36 domain scores were first standardized using a z-score transformation using population norms and metrics for Peruvian women. Each SF-36 domain score was multiplied by the appropriate orthogonal factor scoring coefficient from United States (US) normative data. Resulting scores underwent a t-score transformation to have a mean of 50 and a standard deviation of 10 as is the US norm for international comparison [24, 26]. Individual domain scores were not transformed.

## Data analysis

Descriptive statistics, means, and standard deviations were used for reporting continuous variables, and frequencies and percentages for categorical variables. Parametric tests were used for normally distributed data and nonparametric tests were used for non-normally distributed data. Comparisons between HIV-infected and uninfected women were performed with independent two-tailed t-tests and Mann-Whitney tests for continuous variables as appropriate, and chi-square analysis for categorical variables. Further association between variables and physical activity and HRQoL was assessed using one-way analysis of variance, the Kruskal-Wallis Test, and regression analysis as appropriate. A p-value <0.05 was considered statistically significant.

Sociodemographic and clinical characteristics, along with physical activity scores, were used to fit a multivariable linear regression model with SF-36 component scores as the primary outcome of interest within HIV-infected and noninfected women, separately. Physical activity categorical variables were dichotomized into low activity level and moderate/high activity level. Variables with a p-value <0.25 in the univariate analysis were entered into the multivariable linear regression model and non-significant variables were removed one at a time beginning with the least significant. All variables with a p<0.1 were retained in the model. Age was retained regardless of significance. All statistical analysis was performed using R 3.6.3 and RStudio 1.3.959 with additional packages [27, 28].

## Ethics

This study was reviewed and approved by the institutional review boards of Yale School of Medicine, Universidad Peruana Cayetano Heredia and HNAL. All willing participants provided written informed consent after a comprehensive explanation of the study procedures.

## Results

### Sociodemographic and clinical characteristics

A total of 427 Peruvian women were enrolled in this study, with 175 HIV-infected and 225 uninfected participants. The patients had a mean age of 54±8 (range 40–82) years with post-menopausal women representing 77% of the entire study population (Table 1). HIV-infected women tended to be single/divorced (p<0.001), with incomplete high school (p<0.001), unemployed (p<0.001), and with lower BMI (p<0.01), waist, and hip circumference (p<0.01). Only a minority of the women smoked or consumed alcohol in both groups and these behaviors were not significantly different between groups. IPAQ measurements, which include sedentary scores, total MET scores, and categorical scores, did not vary significantly depending on HIV status. Close to 60% of both groups, however, had a low physical activity categorical score (Table 2).

### SF-36 domains and component scores

Average physical and mental components were lower than the US population norm [26] with transformed physical component average scores at 45.9±8, and mental component average scores at 45.1±6.1 for the HIV-infected women (Table 2). These aggregate scores did not vary significantly depending on HIV status, although the physical component had a p-value of 0.053. For individual SF-36 domains, participant scores were on average highest for "physical functioning," "role limitation due to emotional health," and "pain," and on average lowest for "general health" and "energy/fatigue." Four of the domains: "physical functioning," "role limitations due to physical health," "role limitations due to emotional health," and "emotional wellbeing" were significantly higher for the uninfected compared to the HIV-infected subset.

### SF-36 component summary scores in uninfected women

Unadjusted analysis for the physical component revealed that age, BMI, and postmenopausal status were associated with lower physical scores, and current alcohol use and higher education were independently associated with higher physical scores (Table 3). These factors were added to the multivariable regression model, where older age and higher BMI were found to be associated with lower physical scores. Current alcohol use was positively associated with the physical component. For the mental component, in unadjusted analysis, smoking trended toward significance (p = 0.059) for being negatively associated with the mental component. No other covariant were found to be significantly associated with mental scores in both unadjusted and adjusted analysis (Table 4).

### SF-36 component summary scores in HIV-infected women

When we focused on the group of women with HIV, unadjusted analysis revealed that presence of other comorbidities, mestizo ethnicity, and current alcohol use, were independently associated with a lower physical component and moderate/high physical activity with a higher one (Table 3). These factors, as well as age, menopause status, marital status, education level, occupation, AIDS status, naïve HIV viral load, and years since HIV-diagnosis, were added into a multivariable linear regression analysis. In this model, moderate/high physical activity level was associated with a higher physical component while mestizo ethnicity and presence of chronic comorbidities were associated with a lower physical component. In the unadjusted analysis of the mental component: employment, current CD4+ cell count, and moderate/high physical activity were all independently associated with higher mental component scores. These factors, as well as AIDS status, menopause status, age, and height, were added into a

**Table 1. Sociodemographic and clinical characteristics by HIV status.**

| Characteristics | HIV Status | |
|---|---|---|
| | HIV-infected (N = 175) | Uninfected (N = 252) |
| **Sociodemographic and Behavioral Characteristics** | | |
| Age *Mean ± SD* | **51.32 ± 8.13** | **56.09 ± 8.861**[***] |
| Ethnicity *N (%)* | | |
| White | **7 (4)** | **26 (10)**[**] |
| Mestizo | 162 (93) | 222 (88) |
| Other | 6 (3) | 4 (2) |
| Marital Status *N (%)* | | |
| Single/divorced/separated/widowed | **117 (67)** | **117 (46)**[***] |
| Married/cohabitant | 58 (33) | 135 (54) |
| Education Level *N (%)* | | |
| Incomplete High School | **54 (31)** | **38 (15)**[***] |
| Completed High School | 121 (69) | 214 (85) |
| Occupation status N(%) | | |
| Employed | **4 (2)** | **80 (32)**[***] |
| Unemployed | 171 (98) | 172 (68) |
| **Clinical Characteristics** | | |
| Body mass index *Mean ± SD* | **26.47 ± 5.10** | **27.55 ± 4.022**[**] |
| Height *Mean ± SD* | 154.23 ± 6.08 | 154.07 ± 6.14 |
| Weight *Mean ± SD* | **62.98 ± 12.76** | **65.43 ± 10.542**[**] |
| Hip Circumference *Mean ± SD* | **96.41 ± 11.28** | **100.29 ± 10.791**[**] |
| Waist Circumference *Mean ± SD* | **88.78 ± 11.04** | **92.39 ± 10.631**[**] |
| Smoking, ever *N(%)* | 49 (28) | 50 (20) |
| Current alcohol use *N (%)* | 45 (26) | 66 (26) |
| Menopausal *N (%)* | **123 (70)** | **206 (82)**[**] |
| Diagnosis of other chronic comorbidity *N (%)* | 47 (26) | 118 (46) |
| **HIV-Related Characteristics** | | |
| Current CD4+ Cell Count *Median (IQR)* | 592 (403–796) | – |
| Current Viral Load *Median (IQR)* | 0 (0–40) | – |
| ART-Naive HIV viral load *Median (IQR)* | 24081 (399–184000) | – |
| ART-Naive CD4+Cell Count *Median (IQR)* | 240 (101–395) | – |
| Years since HIV diagnosis *Median (IQR)* | 12 (7–15) | – |
| ART Duration in years *Median (IRQ)* | 10 (6–14) | – |
| AIDS progression *N (%)* | 14 (8) | – |
| Viral suppression *N (%)* | 141 (80.6) | |
| **Physical Activity (IPAQ)** | | |
| Sedentary Score *Mean ± SD* | 180.77 ± 117.95 | 192.06 ± 151.01 |
| Total MET Score *Mean ± SD* | 795.51 ± 1379.60 | 844.69 ± 1207.54 |
| Categorical Scores *N (%)* | | |
| Low | 103 (59) | 147 (58) |
| Moderate | 61 (35) | 83 (33) |
| High | 11 (6) | 22 (9) |

SD: standard deviation; ART: antiretroviral treatment; IPAQ: International Physical Activity Questionnaire; MET: Metabolic Equivalent of Task; IQR: Interquartile range; CD4+ cell count measured in cell/mm$^3$; viral load measured in number of copies/mL; AIDS progression: CD4+ cell count <200 cell/mm$^3$; Viral suppression: viral load ≤50 copies/mL

[*] $p < 0.05$

[**] $p < 0.01$

[***] $p < 0.001$

**Table 2. SF-36 outcomes by HIV status.**

| | HIV status | |
|---|---|---|
| **Characteristics** | **HIV-infected (N = 175)** | **Uninfected (N = 252)** |
| **SF-36 domains,** *Mean ± SD* | | |
| Physical Functioning | **82.37 ± 15.64** | **86.71 ± 11.92**[*] |
| Role Limit Physical | **59.00 ± 35.37** | **66.47 ± 32.75**[*] |
| Bodily Pain | 77.01 ± 12.95 | 75.90 ± 12.69 |
| General Health | 48.60 ± 11.33 | 50.50 ± 11.62 |
| Energy/Fatigue | 48.97 ± 10.70 | 50.46 ± 12.71 |
| Social Functioning | 74.64 ± 12.67 | 76.64 ± 14.08 |
| Role Limit Emotional | **86.29 ± 26.55** | **92.33 ± 22.33**[**] |
| Emotional Wellbeing | **58.19 ± 10.36** | **59.54 ± 12.18**[*] |
| **Physical Component Score** | 45.85 ± 8.000 | 47.29 ± 6.73 |
| **Mental Component Score** | 45.13 ± 6.14 | 45.81 ± 6.46 |

SF-36: 36-Item Short Form Survey

SF-36 domains were scored from 0–100

PCS and MCS were scored from 0–100 and scaled to a mean of 50 and standard deviation of 10

[*] $p < 0.05$

[**] $p < 0.01$

[***] $p < 0.001$

multivariable regression model, which revealed that employment and CD4+ cell count was associated with a higher mental component, while moderate/high physical activity was associated with a lower one.

## Discussion

Our study is the first to explore HRQoL among middle-aged and older women with and without HIV using the well-validated SF-36 questionnaire in Peru. While only four out of the eight SF-36 scores were significantly different between the HIV-infected population and the control group, these domains spanned both physical and mental components of HRQoL as defined by the SF-36. HIV-infected women scored lower for both physical and mental components. In addition, multivariable regression adjusting for various factors including age, BMI, and other sociodemographic factors of the HIV-infected subset illustrated that physical activity levels are also correlated with both physical and mental components.

In the overall sample, results for the SF-36 domains were generally consistent in terms of domain scoring with prior studies conducted using the SF-36 in other Latin American individuals, including a general population study among 4,344 participants in Peru, a validation study on 2,143 older (60–92 years) community-dwelling people in Chile, and a study among 118 HIV-infected individuals in Venezuela [22, 24, 29], all of which were cross-sectional, multicenter studies, but not limited to just women. Aside from two domains, the relationships identified between domains scores within our study were also observed in those Latin American studies. For example, domain scores like "physical functioning" and "social functioning" were all relatively high, while "general health" and "energy/fatigue" were relatively low compared to other scores, a pattern that was consistent across all studies. The two domains that differed in our study compared to the others were "role limitations due to physical health" and "role limitations due to emotional health." Both included a total of seven items (three for the former and

**Table 3. Unadjusted regression of factors related to SF-36 component scores.**

| Independent Variables | Physical Component Score | | | | Mental Component Score | | | |
|---|---|---|---|---|---|---|---|---|
| | Uninfected women | | HIV-infected women | | Uninfected women | | HIV-Infected women | |
| | β | 95% CI | β | 95% CI | β | 95% CI | β | 95% CI |
| **Sociodemographic, Behavioral Characteristics** | | | | | | | | |
| Ethnicity | | | | | | | | |
| white | Ref | - | Ref | - | Ref | - | Ref | - |
| mestizo | 1.2027 | -1.547 to 3.953 | **-8.4608** | **-14.459 to -2.463**** | 0.5370 | -2.101 to 3.175 | 2.0501 | -2.611 to 6.712 |
| other | 3.8383 | -3.287 to 10.964 | -7.8581 | -16.502 to 0.786 | -3.8900 | -10.725 to 2.944 | -2.1098 | -8.828 to 4.608 |
| Marital Status | | | | | | | | |
| single/alone | Ref | - | Ref | - | Ref | - | Ref | - |
| couple | -0.0870 | -1.764 to 1.590 | -1.8393 | -4.367 to 0.689 | -0.0342 | -1.645 to 1.576 | 0.9999 | -0.946 to 2.946 |
| Education | | | | | | | | |
| incomplete high school | Ref | - | Ref | - | Ref | - | Ref | - |
| complete high school | **2.3810** | **0.062 to 4.699*** | 1.8828 | -0.694 to 4.459 | 0.6485 | -1.594 to 2.891 | 0.7406 | -1.246 to 2.727 |
| Occupation | | | | | | | | |
| unemployed | Ref | - | Ref | - | Ref | - | Ref | - |
| employed | -0.9410 | -2.734 to 0.852 | -7.0368 | -14.977 to 0.904 | -0.6846 | -2.408 to 1.039 | **6.4317** | **0.360 to 12.504*** |
| Current alcohol use | | | | | | | | |
| no | Ref | - | Ref | - | Ref | - | Ref | - |
| yes | **2.6063** | **0.732 to 4.481**** | **3.0271** | **0.326 to 5.728*** | -0.3254 | -2.152 to 1.501 | -1.0053 | -3.102 to 1.092 |
| Smoking, ever | | | | | | | | |
| no | Ref | - | Ref | - | Ref | - | Ref | - |
| yes | 1.9562 | -0.127 to 4.039 | 0.8200 | -1.843 to 3.483 | -1.9297 | -3.929 to 0.070 | -0.3759 | -2.422 to 1.670 |
| **Clinical Characteristics** | | | | | | | | |
| Age | **-0.1956** | **-0.287 to -0.104**** | -0.1367 | -0.283 to 0.010 | 0.0765 | -0.014 to 0.167 | 0.0524 | -0.061 to 0.166 |
| BMI | **-0.3158** | **-0.521 to -0.111**** | -0.0961 | -0.331 to 0.139 | 0.0482 | -0.152 to 0.248 | 0.0060 | -0.175 to 0.187 |
| Hip Circumference | -0.0547 | -0.132 to 0.023 | 0.0190 | -0.087 to 0.125 | -0.0027 | -0.077 to 0.072 | -0.0167 | -0.098 to 0.065 |
| Waist Circumference | -0.0733 | -0.152 to 0.005 | 0.0051 | -0.104 to 0.114 | 0.0048 | -0.071 to 0.081 | -0.0231 | -0.107 to 0.060 |
| Menopausal | | | | | | | | |
| no | Ref | - | Ref | - | Ref | - | Ref | - |
| yes | **-3.4560** | **-5.578 to -1.333**** | -1.5487 | -4.158 to 1.061 | 1.4743 | -0.597 to 3.545 | 1.1972 | -0.805 to 3.200 |
| Other Chronic Comorbidities | | | | | | | | |
| no | Ref | - | Ref | - | Ref | - | Ref | - |
| yes | -1.4257 | -3.094 to 0.243 | **-4.0818** | **-6.732 to -1.432**** | -0.5536 | -2.163 to 1.056 | 0.6850 | -1.400 to 2.770 |
| **HIV Clinical Characteristics** | | | | | | | | |
| AIDS progression | | | | | | | | |
| no | - | - | Ref | - | - | - | Ref | - |
| yes | - | - | -3.0460 | -7.435 to 1.343 | - | - | -2.1652 | -5.537 to 1.206 |
| Current CD4+ Cell Count | - | - | 0.0019 | -0.002 to 0.006 | - | - | **0.0034** | **0.0002 to 0.007*** |
| Naive HIV Viral load | - | - | 3.23E-6 | 6.58E-6 to 1.08E-7 | - | - | 4.64E-7 | 3.06E-6 to 2.13E-6 |
| Naive CD4+ Cell Count | - | - | 0.0004 | -0.006 to 0.007 | - | - | -0.0043 | -0.009 to 0.0003 |
| Years since HIV diagnosis | - | - | 0.1694 | -0.033 to 0.372 | - | - | -0.0390 | -0.195 to 0.117 |
| ART Duration (Years) | - | - | 0.1076 | -0.132 to 0.347 | - | - | 0.0795 | -0.104 to 0.263 |
| **Physical Activity[a]** | | | | | | | | |
| Sedentary Score | 0.0036 | -0.002 to 0.010 | -0.0079 | -0.018 to 0.002 | -0.0027 | -0.008 to 0.003 | -0.0004 | -0.008 to 0.007 |
| Total MET Score | -8.76E-05 | -0.00078 to 0.0006 | 0.0001 | -0.001 to 0.001 | -2.54E-05 | -0.0007 to 0.0006 | 0.0002 | -0.0004 to 0.001 |
| Categorical Score | | | | | | | | |
| low | Ref | - | Ref | - | Ref | - | Ref | - |

*(Continued)*

**Table 3.** (Continued)

| Independent Variables | Physical Component Score | | | | Mental Component Score | | | |
|---|---|---|---|---|---|---|---|---|
| | Uninfected women | | HIV-infected women | | Uninfected women | | HIV-Infected women | |
| | β | 95% CI | β | 95% CI | β | 95% CI | β | 95% CI |
| moderate or high | 1.3777 | -0.310 to 3.066 | **3.1382** | **0.752 to 5.525**** | -0.4193 | -2.048 to 1.209 | **-2.1451** | **-3.984 to -0.306*** |

[a]Measured by the International Physical Activity Questionnaire (IPAQ); PCS, physical component score; MCS, mental component score; BMI, body mass index; ART, antiretroviral therapy; CI, confidence intervals

* $p < 0.05$

** $p < 0.01$

*** $p < 0.001$

four from the latter) that explored functional limitation of work-related and daily activities because of physical and emotional problems, respectively. "Role limitations due to physical health" domain scores were lower and "role limitations due to emotional health" scores were higher than what was observed in the aforementioned reports. For physical health role limitations, these lower scores may be likely due to our specific focus on aging women, of which one-third were HIV-positive. For emotional health role limitations, the higher scores among

**Table 4. Adjusted multiple linear regression of factors related to SF-36 component scores.**

| Independent Variables | Physical Component Score | | | | Mental Component Score | | | |
|---|---|---|---|---|---|---|---|---|
| | Uninfected women | | HIV-infected women | | Uninfected women | | HIV-Infected women | |
| | β | 95% CI | β | 95% CI | β | 95% CI | β | 95% CI |
| **Intercept** | **65.95** | **58.33 to 73.57**** | **65.34** | **53.354 to 77.320**** | **41.81** | **36.702 to 46.922**** | **37.3028** | **2.905 to 45.558**** |
| **Sociodemographic, Behavioral Characteristics** | | | | | | | | |
| Ethnicity—White | - | - | Ref | - | - | - | Ref | - |
| Ethnicity—Mestizo | - | - | **-8.8753** | **-14.598 to -3.153**** | - | - | - | - |
| Ethnicity—Other | - | - | -8.1751 | -16.429 to 0.079 | - | - | - | - |
| Employed | - | - | -6.6103 | -14.149 to 0.928 | - | - | **6.8997** | **0.923 to 12.876*** |
| Smoking, ever | - | - | | | -1.9617 | -3.954 to 0.030 | | |
| Current Alcohol Use | **2.4049** | **0.594 to 4.216**** | - | - | - | - | - | - |
| **Clinical Characteristics** | | | | | | | | |
| Age | **-0.1864** | **-0.276 to -0.097**** | -0.0911 | -0.235 to 0.053 | 0.0781 | -0.012 to 0.170 | 0.0009 | -0.113 to 0.115 |
| BMI | **-0.3522** | **-0.548 to -0.157**** | - | - | - | - | - | - |
| Presence of another comorbidity | - | - | **-3.7413** | **-6.297 to -1.186**** | - | - | - | - |
| **HIV Clinical Characteristics** | | | | | | | | |
| Current CD4+ Count | - | - | - | - | - | - | **0.0031** | **0.0001 to 0.006*** |
| **Physical Activity**[a] | | | | | | | | |
| Sedentary Score | 0.0045 | -0.001 to 0.010 | | | - | - | | |
| Categorical Score—Low | - | - | Ref | - | - | - | Ref | - |
| Categorical Score—Moderate or High | - | - | **2.7540** | **0.394 to 5.115**** | - | - | **-2.0919** | **-3.959 to -0.225*** |
| **R² value** | 0.124 | | 0.124 | | 0.018 | | 0.057 | |

[a]Measured by the International Physical Activity Questionnaire (IPAQ); PCS, physical component score; MCS, mental component score; BMI, body mass index; ART, antiretroviral therapy; CI, confidence intervals

* $p < 0.05$

** $p < 0.01$

*** $p < 0.001$

women with HIV may reflect their engagement in longitudinal care at the HIV clinic and resultant regular access and communication with a healthcare team. It will be important however to clearly assess these potential relationships in future longitudinal studies among women with HIV from this region. Finally, compared to the population of a Brazilian monocentric study of 1,057 older woman (mean age 67.1 years) using the SF-36 among women with and without fractures and obesity [30], both HIV-infected and uninfected women from our sample scored lower in both the physical and mental component scores compared to the control group (non-obese without fracture) from that study. This population also followed the pattern of "physical functioning" and "social functioning" being relatively high and "general health" and "energy fatigue" being relatively low compared to the other scores.

Compared to other studies examining HRQoL between individuals with and without HIV using the SF-36, our results were similarly mixed. In our study, scores for four out of the eight domains (physical functioning, role limitations due to physical health, role limitations due to emotional health, emotional wellbeing) were significantly lower for the HIV-infected group than the control. While the general consensus is that PLWH have lower HRQoL scores across all domains [31–33], individual studies have found this to not be the case universally. For example, a study in Japan found "physical functioning" and "bodily pain" scores to be higher in PLWH [34], which represent a better HRQoL in those domains, and a study in the US found "emotional wellbeing" to be unaffected by HIV status [35]. As none of the aforementioned studies were conducted in Latin America nor specifically focused on women, possible variations in HIV treatment and socio-cultural nuances likely influenced the outcomes.

When controlling for other factors through regression analysis, age and BMI were independent predictors of a lower physical component among the uninfected women, which have been noted previously across many studies using the SF-36 [4, 34, 36]. However, for the HIV-infected subset, other factors such as mestizo ethnicity and presence of other comorbidities were independently associated with lower physical scores. These results suggest that despite progress made in the treatment for HIV, for aging Peruvian women, HIV continues to negatively impact the physical aspect of their quality of life and exacerbate other factors like ethnicity and illness. Ethnicity in particular is relevant because of the close association of socioeconomic stratification with racial hierarchies in Peru, of which the most disadvantaged are the indigenous, followed by mestizo ethnic groups [37].

For the mental component model, no factors measured were significant contributors for the uninfected women physical and mental outcomes. However, for the HIV-infected women, employment and higher CD4+ cell count were both significantly associated with a higher mental score. Other studies have reported depression, psychoactive drug usage, and social support as being among the variables correlated with the mental component [4, 34, 36]. As we did not measure these in our study, it is possible that these factors may also contribute to the mental component of aging Peruvian women, and we should not exclude them in future evaluations of HRQoL. It is well known that in general, PLWH can face stigma, prejudice, and discrimination [34, 38]. Thus, it is both expected and commonly observed that HIV diagnosis significantly influences this component. However, even though mental scores were slightly lower for the HIV-infected women compared to the uninfected, the difference was not statistically significant. Factors such as social support have been shown to be positively correlated with the mental component and factors such as stigma have been shown to be negatively correlated with this component [39]. For example, a Colombian cross-sectional study among 286 PLWH found a direct relationship between the emotional well-being dimension of the SF-36 and social support systems assessed by the Duke-UNC-11 questionnaire [39]. The fact that mental component scores were similar between the groups suggests that this sample of middle-aged and older Peruvian women with HIV may not encounter as much stigma or may have the

support that they need to overcome the negative mental health consequences identified among other populations with HIV. However, as this was just one single-center, cross-sectional study, further investigations in this population are needed to confirm this finding.

Given our focus on women aging with HIV, we conducted additional regression models specifically for the HIV-infected subset to evaluate associations with HIV-related clinical characteristics. The presence of comorbidities contributing negatively to the physical component, and employment and current CD4 count contributing positively to the mental component is both expected and confirmed through other studies [35, 36, 40, 41]. Moderate-to-high physical activity contributed positively to the physical component, but also contributed negatively to the mental component, which reflects the mixed reports of other studies. While many reports describe the positive influence of physical activity on HRQoL [42–45], we also found studies that have observed the opposite [46]. A 2019 study in the US using a supervised exercise intervention on 32 PLWH and 37 controls recruited by invitation from an HIV-clinic and referrals from participants, respectively, failed to improve HRQoL measures and depressive symptoms in older PLWH, which suggested this could be related to assignment versus self-selection of the intervention, a greater burden of mental illness in their group, and a longer duration of study. It is important to note that these findings should be interpreted with caution due to the acknowledged limitations of a small sample size and large standard deviations [47]. Another study of 99 Canadian PWLH on psychological and behavioral variables in HRQoL also acknowledged the varied effectiveness of physical activity on HRQoL and attributed it to unequal matching between physical activity intensity and patient tolerance [36]. One possibility for our population is that physical activity levels are comprised of not only intentional exercise, but also unintentional physical activity such as manual labor. Physical health may be positively influenced by both types of physical activity, but mental health may be negatively affected by undesired physical activity.

Our study has some limitations. First, our findings cannot necessarily be extrapolated to represent HIV-infected women outside of the study sample, given it is a single-center study. However, we selected our study site because it is one of the largest public HIV clinics in the capital city of Peru, with a diverse patient population. However, as this study aimed to explore health-related perceptions, it required direct input from the population of interest. Second, we recruited both HIV-infected and uninfected women by convenience sampling because of feasibility reasons, this could potentially add some selection bias that should be taken into consideration. Finally, as this was a cross-sectional study, we were not able to infer causality within the relationships addressed. We hope, however, that it serves as a starting point for future longitudinal research on this subject, especially as it is the first study in Peru that used the SF-36 to measure HRQoL in women and HIV populations.

The SF-36 is a widely adopted tool that has been validated in different languages to measure HRQoL. With psychometric data supporting its usage in many disease areas, SF-36 has been the most frequently used generic HRQoL survey for PLWH, resulting in refined scoring protocols, numerous validation studies across different populations, and ample literature for clarification and comparison in this particular population [2, 9, 48, 49]. In low- and middle-income countries such as Peru, information regarding the performance of self-reported health surveys is important because these measures are comprehensive, practical and inexpensive, making them valuable for health intervention evaluations [19], particularly in the setting of HIV in Latin America, where there is an important pending agenda to address gaps in information regarding long-term care of PLWH [12]. More specifically, it is important to address aspects that are intrinsic to the patient-perspective, which HRQoL measures can help elucidate. Using findings from HRQoL studies can thus allow for better implementation on the individual and communitarian level.

As the number of HIV-infected women on ART continues to increase in Latin America, it is important to acknowledge that improvements in life expectancy because of ART have also led to new challenges in terms of the physical and mental burdens associated with life-long infection and HIV treatment. In summary, although overall physical and mental component scores were similar between the two groups, we found that HIV-infected women scored lower in specific domains within physical and mental health compared to uninfected women, consistent with our hypothesis that HIV status may be negatively associated with HRQoL among Peruvian women. In addition, while physical activity affected both physical and mental status in the HIV-infected subset, it did so in opposite ways. Directions for future studies could include how social support and HIV stigma affect HRQoL in women with HIV in Peru, and further elucidation of the relationship between physical activity and HRQoL in PLWH to determine how different types of physical activity, such as duration, frequency, and purpose, may affect SF-36 component scores. Moreover, our study also identified certain areas in which women with HIV in Peru appear to be thriving relative to reports from the literature, which illuminates opportunities for future research that examines resiliency factors that could ultimately benefit other patients with HIV. This study is the first that we know of to provide insights into HRQoL outcomes among HIV-infected and uninfected women in Peru. Better understanding of the influence of sociodemographic, behavioral, clinical, and activity characteristics of this population on HRQoL will help guide future interventions, policies, and practices that could improve HRQoL of women aging with HIV in Peru.

## Author Contributions

**Conceptualization:** Diego M. Cabrera, Meibin Chen, Mijahil P. Cornejo, Yvett Pinedo, Patricia J. Garcia, Evelyn Hsieh.

**Data curation:** Diego M. Cabrera, Meibin Chen, Mijahil P. Cornejo, Patricia J. Garcia, Evelyn Hsieh.

**Formal analysis:** Diego M. Cabrera, Meibin Chen, Mijahil P. Cornejo, Patricia J. Garcia, Evelyn Hsieh.

**Funding acquisition:** Diego M. Cabrera, Patricia J. Garcia, Evelyn Hsieh.

**Investigation:** Diego M. Cabrera, Mijahil P. Cornejo, Yvett Pinedo, Patricia J. Garcia, Evelyn Hsieh.

**Methodology:** Diego M. Cabrera, Meibin Chen, Mijahil P. Cornejo, Yvett Pinedo, Patricia J. Garcia, Evelyn Hsieh.

**Project administration:** Diego M. Cabrera, Mijahil P. Cornejo, Yvett Pinedo, Patricia J. Garcia, Evelyn Hsieh.

**Resources:** Diego M. Cabrera, Yvett Pinedo, Patricia J. Garcia, Evelyn Hsieh.

**Software:** Diego M. Cabrera, Meibin Chen, Evelyn Hsieh.

**Supervision:** Diego M. Cabrera, Yvett Pinedo, Patricia J. Garcia, Evelyn Hsieh.

**Writing – original draft:** Diego M. Cabrera, Meibin Chen, Mijahil P. Cornejo, Yvett Pinedo, Patricia J. Garcia, Evelyn Hsieh.

**Writing – review & editing:** Diego M. Cabrera, Meibin Chen, Mijahil P. Cornejo, Yvett Pinedo, Patricia J. Garcia, Evelyn Hsieh.

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
