## [Decision Letter · Decision Letter 0]

6 Oct 2021

PONE-D-21-11937Health-Related Quality of Life among Women Aging with and without HIV in Peru

PLOS ONE

Dear Dr. Hsieh,

Thank you for submitting your manuscript to PLOS ONE. After careful consideration, we feel that it has merit but does not fully meet PLOS ONE’s publication criteria as it currently stands. Therefore, we invite you to submit a revised version of the manuscript that addresses the points raised during the review process.

The manuscript has been evaluated by three reviewers, and their comments are available below.

The reviewers have raised a number of concerns that need attention. They request additional information or make queries regarding: the presentation of statistics for a wider readership, general data presentation, study limitations, study context, methodology, discussion points and more.

Could you please revise the manuscript to carefully address the concerns raised?

We look forward to receiving your revised manuscript.

Kind regards,

Sebastian Shepherd

Associate Editor

PLOS ONE

Journal Requirements:

5. Please note that in order to use the direct billing option the corresponding author must be affiliated with the chosen institute. Please either amend your manuscript to change the affiliation or corresponding author, or email us at plosone@plos.org with a request to remove this option.

Reviewers' comments:

Reviewer's Responses to Questions

**Comments to the Author**

1. Is the manuscript technically sound, and do the data support the conclusions?

Reviewer #1: Yes

Reviewer #2: Yes

Reviewer #3: Yes

2. Has the statistical analysis been performed appropriately and rigorously? 

Reviewer #1: Yes

Reviewer #2: Yes

Reviewer #3: Yes

3. Have the authors made all data underlying the findings in their manuscript fully available?

Reviewer #1: Yes

Reviewer #2: No

Reviewer #3: No

4. Is the manuscript presented in an intelligible fashion and written in standard English?

Reviewer #1: Yes

Reviewer #2: Yes

Reviewer #3: Yes

5. Review Comments to the Author

Reviewer #1: This is a very important paper on physical activity of women living with HIV.

SF-36 PCS and MCS scores transformation to have a mean of 50 and standard deviation of 10 should be better described for readers not familiarized with SF-36.

Alcohol and smoking status dichotomous assessment represents a limitation.

The authors should explore more the association of physical activity; it has affected both physical and mental status in the HIV-infected subset in opposite directions. What hypothesis could be formulated?

In the limitations paragraph the authors states that “findings cannot be extrapolated to represent HIV-infected women outside of Lima”. It should be rephrased. It lacks external validity even for HIV-infected women living in Lima , since it was not a probabilistic sample and was an hospital-based study.

Is there any plan to use SF-36 data in further health economic evaluations of technologies to improve physical activity for this population?

Reviewer #2: The manuscript “Health-Related Quality of Life among Women Aging with and without HIV in Peru” compares health related quality of life (measured by the SF-36) among aging women living with and without HIV and predictors of health related quality of life among these two samples. There are some interesting and surprising findings and I commend authors for exploring these issues among a population that is rarely studied. There were a few areas that, if addressed, would enhance the manuscript:

• Introduction

o It seems like the introduction could be improved by including directional hypotheses. Did authors predict that women living with HIV would have worse health related quality of life than those without HIV? Why did they predict this?

o Additionally, it seems like there are some important contextual factors missing from the examination of health-related quality of life. For example, what would the literature suggest about the impact of illness, stigma, etc. on health in this population? What are important contextual factors specific to Peru? Are there specific policies in Peru? Histories of stigma? Impact of religious stigma? Etc. Even a few sentences might “set up” why this is an important question in this location (other than that it has not been studied before). Authors point to the “unequal status of women” – how is that related to sexual health empowerment or lack thereof in Peru? How does that impact the context in which living with HIV is experienced?

• Methods

o Smoking history as “ever vs. never” seems less important to me than “current vs. not current.” Within the U.S. people living with HIV are much more likely to smoke cigarettes than the general population. Is this the same for Peru? It would have been nice to know whether individuals are current smokers, and this likely would be much more highly correlated with current health and health related quality of life than “ever” smoking status. This is particularly true because of the age of individuals – many may have smoked when they were younger and have since stopped. Without this information (or even years of smoking) the impact of this health behavior is less meaningful.

o Although the Spanish version of the SF-36 was used, I wonder if authors could comment on the cultural appropriateness of answering these questions within a medical setting in Peru. During pilot testing, did any patients comment on answering these questions? Did they seem comfortable giving a range of responses? I’m wondering if there is a chance that people may have answered in a socially desirable way because of cultural expectations or norms about health vs. illness. While outside the scope of this paper it might be worth commenting on the appropriateness of the use of this measure within the methods and/or acknowledging potential limitations in the discussion.

• Results

o Time since diagnosis and ART duration are presented as mean values with standard deviations. It would have also been nice to see the range of values for the sample and perhaps even the median value.

o HIV-related health values like CD4+ cell count and nadir and current viral load should also present median values. Mean values could be misrepresentative if there are extreme outliers.

o Current viral load mean seems extremely high for a population engaged in care. What could explain viral loads of 21,000+? This relates to my prior comment about presenting median values instead of means as this may be skewed by high outlier values. Typically, log viral load is what is presented in research – is that what these values are? Or have they not been converted to log values yet? Additionally, it would be beneficial to see how many were considered undetectable (different definitions by generally, <200 has been used internationally, <20 log viral load is used in most clinical settings in the U.S.) versus having a detectable viral load. It would make sense to look at the impact of viral load in this way as well.

• Discussion

o There is a statement made about “sociodemographics contributing to health” because mestizo-identified women had worse health than white identified women. However, I think this is better reframed as mestizo-identified women experienced worse self-reported health likely as a result of social forces and experiences. It might be worth describing how race and social status, health, etc. are related in general, within the global South, and/or within Peru.

o In general, there were some surprising findings that would benefit from additional post-hoc explanations and references to literature. For example, current alcohol use was associated with better health among women living with HIV and women without HIV. Why do authors think this is true? One potential explanation that came to mind is that it is possible individuals who drink alcohol are also socializing more, and that social support often promotes better health. Authors should explore this and other potential explanations for some of there findings rather than just leaving them.

o Another surprising finding was that women living with HIV reported fewer role limitations due to physical or emotional health. Why is this? Is it that they are employing perspective taking? Are happy about their health considering their HIV diagnoses? Could it be that they are more highly engaged in caring for their health because of HIV? That they seek medical care more often through an infectious disease clinic? Authors do not need to know the reasons for these surprising findings, but I do believe it would be beneficial to explore potential explanations in the context of plausible conclusions based on their knowledge of the sociocultural context, literature, etc.

o Within the limitations, authors state that their findings cannot be extrapolated or generalized outside of the specific city of Lima. Why is this? Is the sample in Lima not representative of Peru in general?

o In the final paragraph, authors acknowledge potential next steps in research, however, they are missing one somewhat obvious next step which is to understand how women living with HIV in Peru are thriving. This sample seems to report better health than those without HIV and maybe there are mental, social, or lifestyle behaviors that if identified could be leveraged to improve the health of women living with HIV. Rather than a deficit focus, this paper nicely lends itself to a resiliency focus of people living with HIV and this should be emphasized within the conclusion and carried throughout to strengthen the findings.

Reviewer #3: Review Report

Title: Health-Related Quality of Life among Women Aging with and without HIV in Peru

Reviewer: Prof. Sam Ibeneme

Institutional affiliations:

1. University of Nigeria, Enugu Campus, Department of Medical Rehabilitation, Faculty of Health Sciences, Enugu, Nigeria. sam.ibeneme@unn.edu.ng.

2. Department of Physiotherapy, Faculty of Health Sciences, School of Therapeutic Studies, University of the Witwatersrand, 7 York Road, Parktown, 2193 Johannesburg, South Africa

Reviewers Report

General comments

Many thanks for inviting me to review the suitability of the article Health-Related Quality of Life among Women Aging with and without HIV in Peru’ for publication. Essentially, this article sought to (1) determine whether the effects of aging on HRQOL vary in aging women with or without HIV infections, and (2) to explore the explanatory variables for the variance in HRQOL in aging women with or without HIV infections using predictive models. Therefore, they compared older adult women with and without HIV infection to know whether the impact of aging on HRQOL is similar in both groups or whether there are differences in both groups which could be accounted for by HIV-related factors. The authors also explored the predictive models that could explain differences in HRQOL between the older adult women with or without HIV infection. The findings seem to demonstrate that HRQOL is more negatively affected among older adult women with HIV than otherwise. The summary score for the physical component of HRQOL in the HIV subset was significantly associated with physical activity, ethnicity, and chronic comorbidities suggesting a role for physical activity, socio-cultural factors and HIV-related/age-related complications in the determination of the physical wellbeing of the aging HIV-infected women. Similarly, the mental component was significantly associated with physical activity, employment, and CD4+ cell count suggesting a role for a physical activity lifestyle, social functioning/economic factors and symptoms of the disease severity in the determination of the emotional wellbeing of the aging HIV-infected individuals. Overall, it was shown that HRQOL may have important clinical application in the early identification of the key drivers of physical and emotional welling in the aging HIV infected individuals. These drivers should be prioritised in the clinical management and policy interventions of HIV disease which may vary across socio-cultural boundaries considering the impact of ethnicity. Thus, I consider this study to be original, useful, timely and relevant.

Specific Comments

Following a preliminary review of this article, I find this paper quite interesting and an important scientific contribution towards understanding how the effects of aging on HRQOL vary in individuals with or without HIV infections, and likewise in identifying the explanatory variables that account for these differences. However, there are specific issues that need to be addressed to improve the quality of this paper.

Abstract

Background: The aim of this study is not stated in the background, rather a feeble attempt was made to provide the purpose of this study in the result section. Therefore, these sections of the abstract must be revisited. It was not indicated why the study focused on women and not both sexes.

Method: The sampling technique used should be indicated and likewise the level at which alpha was set. It must be indicated whether the p-value was one-tailed or two-tailed to give proper context when interpreting it. The explanatory variables that were explored in the regression analysis must be mentioned.

Conclusion: The authors reported that “the physical component was significantly associated with physical activity, ethnicity, and chronic comorbidities while the mental component was significantly associated with physical activity, employment, and CD4+ cell count.” However, they did not mention the implication of these relationships in the conclusion. For instance, CD4+ count enables the assessment of HIV symptom severity which should have an impact on emotional wellbeing when such symptoms become more severe. Similarly, ethnicity alludes to socio-cultural factors which influence people's lifestyle and way of life. Therefore, the implications of these findings should be given the proper context to appreciate the implications of the findings of this study. It was stated in the conclusion that “Of note, physical activity affected both 47 components in the HIV-infected group, but with opposite associations,” for this reason, the results should at least state the variables with the most significant negative and positive values, and likewise the proportion of variance that accounted for the relationship or the effect size.

Main study

Background:

The background to this study is well written and provides a good rationale for the present study.

Method:

1. The research design suggests that this is an observational cross-sectional study. It must be clearly stated so that there is a distinction bearing in mind that there are other types of cross-sectional study which fulfil other research purposes. Also, since this study was done among women with HIV and not AIDS, the defining cut-off point for the CD4+ cell count must be stated in the inclusion criteria.

2. It was not indicated how the sample size was determined to ensure that it has the power to detect the differences in the variables of interest between the study group and the comparison group.

Measures and data collection

In line 115, page 5, it was stated that a pilot study was conducted but the result of the pilot was not indicated as regards the validity and reliability of the test instrument. The validity and reliability of the test instruments - SF-36, IPAQ - must be indicated especially since it has been validated in Spanish.

Data Analysis

The type of t-test statistic that was applied in the study should be stated – whether it was independent or the dependent t-test. It is important to indicate whether the p is one-tailed or two-tailed. This is important because this study is driven by a non-directional hypothesis in which case a two-tailed p-value should be appropriate.

Results

1. The result section was well written.

2. However, the varied participants characteristics between the two groups show that they were not equivalent with respect to the variables that are likely to influence the HRQOL. First, the participants in the comparisons group were significantly older, more employed, of greater: BMI, adiposity, and white ethnic heritage. Given that some of these factors are also variate determinants of HRQOL, an explanatory model of the variance in the HQOL on the basis of other predictive variables may have errors except these covariates are appropriately controlled.

3. The results of the multivariate regression analysis should include the R-value to give a sense of the proportion of the population that accounted for the significant variance change in HQOL using the predictive model.

Discussion

1. The last sentence in paragraph one, page 19, reads as follows:

“As none of the aforementioned studies were conducted in Latin America nor specifically focused on women, a combination of changes in HIV treatment and socio-cultural nuances likely influenced the outcomes.”

2. I suggest that the sentence should be modified by replacing “a combination of changes” with “possible variations” since the authors are not certain of this claim.

3. I do not seem to agree with the interpretation of the findings highlighted in the last two sentences of the second paragraph on page 19 which stated as follows:

“However, for the HIV-infected subset, other factors such as mestizo ethnicity and presence of other comorbidities were independently associated with lower physical scores. These results suggest that despite progress made in the treatment for HIV, for aging Peruvian women, HIV continues to negatively impact the physical and sociodemographic aspects of their quality of life.”

In my view, the results indicated that those infected with HIV, and who were of mestizo ethnicity, were at a greater risk of poor physical functioning or physical health state than the rest. These hints of possible health disparity across ethnic boundaries that limit physical functioning in older women with HIV infection in Peru. However, given that such a relationship was not established among the control group, does not mean that it does not exist. Rather it could imply that HIV infection amplified some health disparities across socio-cultural boundaries in older women in Peru, in a manner that placed the mestizo ethnicity at a disadvantage. Since physical health in the HIV-infected population is also dependent on comorbidities, it implies that whatever factors that promote the onset of these comorbidities could be averse to physical health. Such factors may include limited access to healthcare, aging, prolonged usage of ART, socio-economic or employment (economic) inequality among others. Therefore, it could be investigated whether HIV-comorbidities are also common in the mestizo ethnicity than other ethnic groups and whether a similar trend was observed in studies done in other multi-ethnic countries in Latin America with similar access to health for all the ethnic groups as obtained in Peru.

4. The authors reported that physical activity has a mixed impact on the components of HRQOL since the emotional component was not improved in their study, and cited Goulding et al., 2019 (ref. 43). However, it would have been appropriate to also mention that the findings of Goulding et al., (2019) should be interpreted with caution because of the acknowledged limitations in the small sample size and large standard deviations recorded in the measured outcomes, which suggest that there was a wide margin of error or some degree of imprecision in the measurement. However, the authors had also mentioned that:

“In addition, multivariable regression adjusting for various factors including age, BMI, and other sociodemographic factors of the HIV-infected subset illustrated that physical activity levels are also correlated with both physical and mental components.”

Invariably, physical activity or exercise should be expected to influence physical and emotional HRQOL so far as the influence of age, BMI and socio-demographic factors are eliminated or controlled. Therefore, age, BMI and socio-demographic factors are confounding variables that may minimise the effects of an exercise intervention on HRQOL, and ought to be considered when interpreting the results of similar studies.

5. The authors in the introduction (see page 2, line 95), “hypothesized that HIV burden may be an important predictor for physical and mental health of this population.” However, it is not discussed whether this hypothesis was realised, and should be reflected in the discussion.

Conclusion

The conclusion section is meant to reflect on the outcome of the study concerning the set study objectives. Anything outside of this may distract from a primary need to project the key findings of the present study. I am wondering what the key objectives of this study are and whether they were met. I suggest that the specific objectives are clearly stated. Also, I do not find where a mention was made of whether the study hypothesis was accepted or rejected based on the findings.

Orthographic check

There is a need for minor orthographic check of the manuscript.

Other Comments

This is a novel study which provided scientific findings that may have useful application in the clinical management and policy interventions of HIV disease.

Missing sections

The following sections are missing at the back end of this manuscript and should be provided by the authors in line with the journal format and style, including:

- List of Abbreviations

- Ethics approval and consent to participate

- Consent to publish

- Availability of data and materials

- Competing interests

- Funding

- Authors' Contributions

-

6. PLOS authors have the option to publish the peer review history of their article (what does this mean?). If published, this will include your full peer review and any attached files.

Reviewer #1: No

Reviewer #2: No

Reviewer #3: **Yes: **Prof Sam Chid Ibeneme

---

## [Author Response · Author response to Decision Letter 0]

22 Dec 2021

We have prepared a "Response to Reviewers" document addressing all the suggestions and comments made by the editor and the three reviewers. Please refer to that document for specific responses.

---

## [Decision Letter · Decision Letter 1]

13 May 2022

Health-Related Quality of Life among Women Aging with and without HIV in Peru

PONE-D-21-11937R1

Dear Dr. Hsieh,

We’re pleased to inform you that your manuscript has been judged scientifically suitable for publication and will be formally accepted for publication once it meets all outstanding technical requirements.

Kind regards,

Jerome T Galea, PhD, MSW

Academic Editor

PLOS ONE

Additional Editor Comments (optional):

Reviewers' comments:

Reviewer's Responses to Questions

**Comments to the Author**

1. If the authors have adequately addressed your comments raised in a previous round of review and you feel that this manuscript is now acceptable for publication, you may indicate that here to bypass the “Comments to the Author” section, enter your conflict of interest statement in the “Confidential to Editor” section, and submit your "Accept" recommendation.

Reviewer #1: All comments have been addressed

Reviewer #2: All comments have been addressed

Reviewer #3: All comments have been addressed

2. Is the manuscript technically sound, and do the data support the conclusions?

Reviewer #1: Yes

Reviewer #2: Yes

Reviewer #3: Yes

3. Has the statistical analysis been performed appropriately and rigorously? 

Reviewer #1: Yes

Reviewer #2: Yes

Reviewer #3: Yes

4. Have the authors made all data underlying the findings in their manuscript fully available?

Reviewer #1: Yes

Reviewer #2: Yes

Reviewer #3: Yes

5. Is the manuscript presented in an intelligible fashion and written in standard English?

Reviewer #1: Yes

Reviewer #2: Yes

Reviewer #3: Yes

6. Review Comments to the Author

Reviewer #1: I am happy with the response letter and the new version of the manuscript in which all comments were addressed. It will be a very important paper in its field.

Reviewer #2: The comments have been adequately addressed within the current version of the manuscript. In addition to improving this manuscript, I believe that some of the findings lend themselves nicely to future studies in the same population, which I hope authors will pursue!

Reviewer #3: Reviewers Report

General comments

Many thanks for inviting me to review the revised version of the article: ‘Health-Related Quality of Life among Women Aging with and without HIV in Peru’ - for publication.

Specific Comments

I have gone through the revised manuscript and feel satisfied that the authors reflected all the corrections as required. The quality of this manuscript has greatly improved and is fit for publication.

Recommendation

I recommend that this article is accepted for publication as it is.

7. PLOS authors have the option to publish the peer review history of their article (what does this mean?). If published, this will include your full peer review and any attached files.

Reviewer #1: **Yes: **Rodolfo Castro

Reviewer #2: No

Reviewer #3: **Yes: **Prof. Sam Chidi Ibeneme

---

## [Editor Report · Acceptance letter]

1 Jun 2022

PONE-D-21-11937R1 

Health-Related Quality of Life among Women Aging with and without HIV in Peru 

Dear Dr. Hsieh:

I'm pleased to inform you that your manuscript has been deemed suitable for publication in PLOS ONE. Congratulations! Your manuscript is now with our production department. 

Kind regards, 

on behalf of

Dr. Jerome T Galea 

Academic Editor

PLOS ONE